# A neural mechanism of speed-accuracy tradeoff in macaque area LIP

**Timothy Hanks[1]\*, Roozbeh Kiani[2], Michael N Shadlen[3]\***

[1]Princeton Neuroscience Institute, Princeton University, Princeton, United States; [2]Center for Neural Science, New York University, New York, United States; [3]Department of Neuroscience, Howard Hughes Medical Institute, Columbia University, New York, United States

**Abstract** Decision making often involves a tradeoff between speed and accuracy. Previous studies indicate that neural activity in the lateral intraparietal area (LIP) represents the gradual accumulation of evidence toward a threshold level, or evidence bound, which terminates the decision process. The level of this bound is hypothesized to mediate the speed-accuracy tradeoff. To test this, we recorded from LIP while monkeys performed a motion discrimination task in two speed-accuracy regimes. Surprisingly, the terminating threshold levels of neural activity were similar in both regimes. However, neurons recorded in the faster regime exhibited stronger evidence-independent activation from the beginning of decision formation, effectively reducing the evidence-dependent neural modulation needed for choice commitment. Our results suggest that control of speed vs accuracy may be exerted through changes in decision-related neural activity itself rather than through changes in the threshold applied to such neural activity to terminate a decision.

\*For correspondence: thanks@ princeton.edu (TH); shadlen@ columbia.edu (MNS)

**Competing interests:** The authors declare that no competing interests exist.

**Reviewing editor**: Dora E Angelaki, Baylor College of Medicine, United States

## Introduction

It is well known that the accuracy of many types of decisions suffers when there is pressure to decide quickly. Conversely, decision accuracy often improves if temporal demands permit longer deliberation. This balance between decision time and performance is referred to as the speed-accuracy tradeoff (SAT). Examples of SAT are prevalent, with demonstrations for visual, auditory, olfactory, and memory tasks (*Green and Luce, 1973*; *Reed, 1973*; *Wickelgren, 1977*; *Luce, 1986*; *Reddi and Carpenter, 2000*; *Reddi et al., 2003*; *Palmer et al., 2005*; *Rinberg et al., 2006*; *Ings and Chittka, 2008*; *Bogacz et al., 2010*).

Models based on the stochastic accumulation of evidence to a bound provide a unifying behavioral framework that can explain both decisions and the time taken to reach those decisions (*Stone, 1960*; *Laming, 1968*; *Link, 1992*; *Ratcliff and Rouder, 1998*; *Ratcliff and Smith, 2004*; *Palmer et al., 2005*). In this class of models, the decision bound establishes a policy on the amount of evidence needed for decision commitment and thus determines the tradeoff between speed and accuracy. Providing confirmation of this idea, multiple studies have found that human SAT in a variety of decision tasks can be explained by changes in decision bound (*Reddi and Carpenter, 2000*; *Reddi et al., 2003*; *Palmer et al., 2005*).

While providing insight into computation, these mathematical models of behavior leave open the question of neural implementation. Neuroimaging studies suggest that SAT may be implemented by changes in baseline neural activity in decision-related areas, with higher baseline responses when speed is given precedence over accuracy (*Forstmann et al., 2008*; *Ivanoff et al., 2008*; *van Veen et al., 2008*; *Forstmann et al., 2010*; *Wenzlaff et al., 2011*). This suggests the possibility that a reduction of decision bound in behavioral models is implemented in the brain by increasing the starting level of neural activity in regions that accumulate evidence rather than by reducing the bound

**eLife digest** Many actions involve a trade-off between speed and accuracy, with typing being a good example: the faster you try to type a sentence, the more mistakes you are likely to make. Mathematical models have successfully reproduced the speed-accuracy trade-off, but it is not clear how the brain represents and weighs up these two factors. Now, Hanks et al. have shown how single neurons in a region of the brain called the lateral intraparietal cortex vary their firing rate to optimize the balance between speed and accuracy.

Two macaque monkeys were trained to fixate on a single dot on a screen and then move their eyes in one of two directions in response to movies of random dots on a video screen. Initially, the monkeys received a reward immediately after every correct response, whereas incorrect responses were punished with a very short time-out. Under these conditions, the optimal strategy is to respond quickly at the expense of accuracy. In a separate block of trials, the monkeys were again rewarded for correct responses, but this time their reward was delayed if they responded too quickly. The most effective strategy now is to respond accurately, but more slowly.

In both the 'high speed' and 'high accuracy' conditions, the firing of neurons in lateral intraparietal cortex increased while the dots were on the screen. As soon as the firing rate reached a threshold—representing the point at which the monkey had accumulated enough evidence to make a decision about the direction of movement—the monkey moved its eyes. Previous theories had suggested that when speed was the priority, the level of activity required to trigger a decision would be lower than when accuracy was emphasized. Surprisingly, however, the threshold did not differ between the 'high speed' and 'high accuracy' conditions. Instead, neurons displayed a higher *initial* firing rate whenever speed was prioritized, enabling the monkey to make a decision on the basis of less evidence.

This finding is consistent with human brain imaging studies that have shown increased baseline activity in decision-making circuitry when speed is prioritized over accuracy. Studying these mechanisms could help to reveal why some individuals are more impulsive decision-makers than others.

level required to terminate a decision. The only neurophysiological study of SAT to date reported that during a visual search task with different response deadlines, SAT was associated with a variety of effects in FEF, including changes in the baseline and gain of visual neurons, the duration of perceptual processing, and the rate of ramping responses of movement neurons (*Heitz and Schall, 2012*).

Here we investigate the neural processes that may underlie SAT by recording from the lateral intraparietal area (LIP) of the rhesus monkey during a visual motion discrimination task. We focus on LIP because numerous studies suggest that neurons in this area exhibit the signatures of both evidence accumulation and a decision bound that leads to commitment for saccadic decisions (*Shadlen and Newsome, 2001*; *Roitman and Shadlen, 2002*; *Hanks et al., 2006*; *Yang and Shadlen, 2007*; *Churchland et al., 2008*; *Kiani and Shadlen, 2009*; *Rorie et al., 2010*; *Churchland et al., 2011*; *Bollimunta et al., 2012*). We found that SAT was implemented in LIP by an evidence-independent signal that was added to responses during the same period that the neurons represent the accumulating evidence leading to a choice. The addition of this time-dependent, evidence-independent signal serves to reduce the amount of evidence needed to terminate a decision in the high speed regime.

## Results

Two rhesus monkeys (Macaca mulatta) were trained to perform a reaction time (RT) motion discrimination task in two speed-accuracy regimes (*Figure 1*). The stimulus consisted of a patch of dynamic random dots, with different motion strengths randomly interleaved on a trial-by-trial basis ('Materials and methods'). The task of the subjects was to determine the direction of the coherent motion and to indicate a choice by making an eye movement to a corresponding target. Since this was a RT task, the viewing epoch was not preset, and the monkey controlled when to make a choice on each trial.

We manipulated the monkeys' SAT primarily by controlling the timing of reward delivery. To encourage speed, we simply rewarded the monkey immediately after each correct response, since, in our experience, monkeys are naturally inclined to make fast responses at the expense of accuracy.

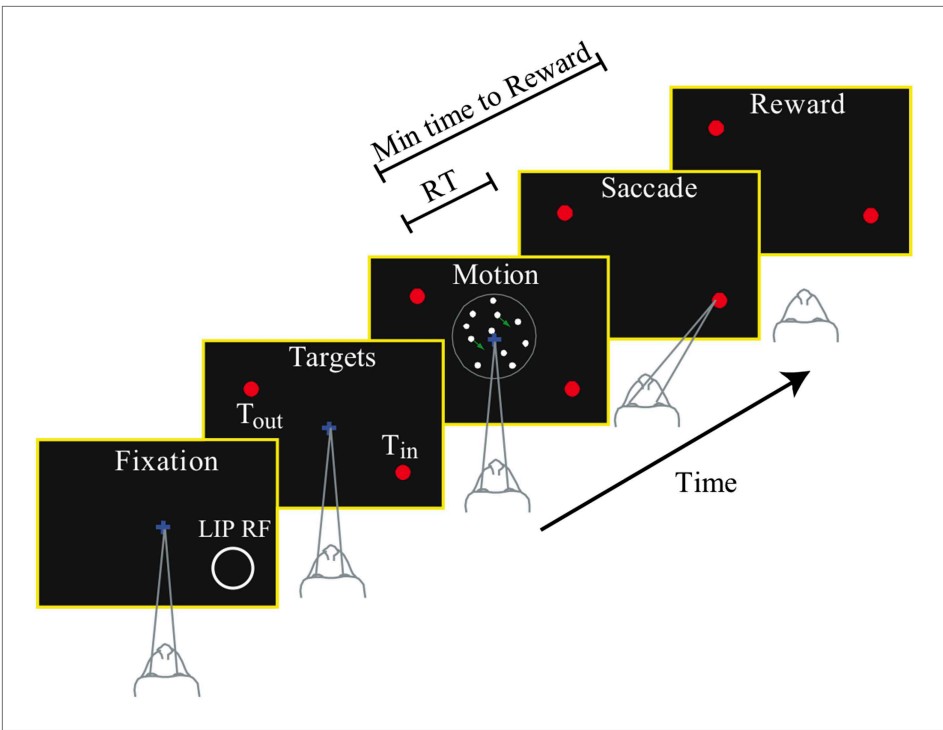

**Figure 1**. Choice-reaction time task. Monkeys were required to identify the net direction of motion in a dynamic random dot display. Several levels of difficulty were randomly interleaved. The monkeys indicated their choice by making a saccadic eye movement to the appropriate choice target. They could do so at any time after onset of the random dot motion stimulus. Reaction time (RT) was measured from the time of motion onset until initiation of the saccade. The monkeys learned to perform the task in two different speed-accuracy regimes. For neural recordings, one choice target was placed in the response field of a single LIP neuron, as illustrated, and the other target was placed in the opposite hemifield. The circle showing the RF is for illustration purposes and was not displayed to the monkey. The target in the neuron's RF is referred to as $T_{in}$; the other target is referred to as $T_{out}$.

To encourage accuracy, reward was delayed relative to motion onset by 600–800 ms, so that fast responses involved a short additional wait until delivery of reward. We refer to the first regime as the 'speed' regime and the second as the 'accuracy' regime. Each speed-accuracy regime was maintained for many consecutive daily sessions. Neural data were collected when the monkey had achieved a stable set point for each regime.

We first characterize the monkeys' behavior in the two speed-accuracy regimes. We show that the change in behavior in the different speed-accuracy regimes is explained by a change in the decision criterion. Second, we turn to a neural correlate of this change. We show that LIP neurons reflect the change in the decision policy as a motion evidence-independent signal that adds to the responses throughout the period of evidence accumulation.

## Behavior

In both regimes, stronger motion was associated with improved choice accuracy and faster RTs, as previously shown (*Figure 2A*; *Roitman and Shadlen, 2002*). Behavior during the two regimes revealed a tradeoff between speed and accuracy. In the high accuracy regime, both monkeys exhibited longer RTs and improved performance (*Figure 2A*). This observation is important because it indicates that the monkeys did not slow down merely by delaying the motor response, but rather, they used the time to accumulate more evidence from the stimulus. Similarly, if the monkeys had failed to pay attention to the early portion of the stimulus, an improvement in accuracy should not have accompanied the longer RTs.

A bounded accumulation model can explain the behavior in both regimes (*Figure 2B*). The traditional depiction of this model (*Figure 2B*) would represent the accumulation of evidence as a random walk between two terminating bounds. We consider a version of this model that is more consistent

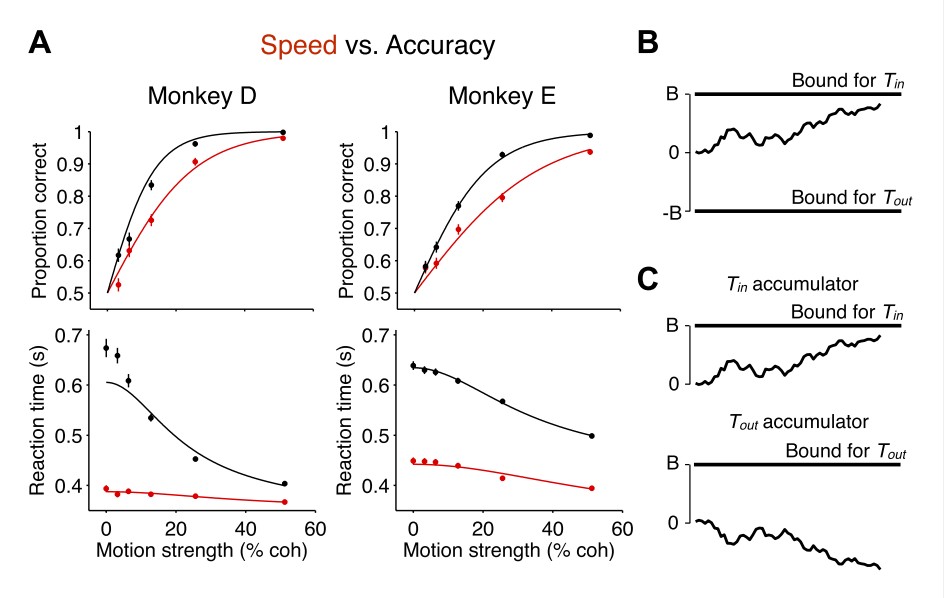

**Figure 2**. Decision accuracy and reaction time in two speed-accuracy regimes are explained by bounded evidence accumulation. (**A**) Data from two monkeys. Black symbols and lines show the high accuracy regime. Red symbols and lines show the high speed regime. The top graph shows the proportion of correct choices plotted as a function of motion strength. Decision accuracy improved with stronger motion in both regimes. The bottom graph shows mean RT (±SEM) plotted as a function of motion strength. RTs decreased for stronger motion in both regimes. In the regime where the monkeys responded more slowly, their performance improved relative to the other regime. N = 3534 and 3307 trials for the high speed and high accuracy regimes, respectively, for monkey D and 4838 and 4733 trials for the same regimes for monkey E. (**B** and **C**) Bounded evidence-accumulation models. Noisy evidence is accumulated until it reaches a terminating threshold or bound, which establishes the time and sign of the decision. (**B**) Traditionally, this is represented as a single accumulation—or drift-diffusion process—between an upper and lower bound. (**C**) The model can be described equivalently as a race between two anticorrelated drift-diffusion processes, such that negative evidence for one process is positive evidence for the other. The decision is determined by the first accumulation to reach the bound. The traces in (**A**) show fits to the data using the bounded accumulation model.

with neurophysiology (*Figure 2C*) in which one process is replaced by two competing processes that accumulate moment-by-moment sensory evidence bearing on each alternative (*Usher and McClelland, 2001*). The choice is dictated by which process first reaches the bound (*Figure 2C*). This two-process model explains the established neural representations of both saccadic choices in LIP for this task (*Mazurek et al., 2003*). In the model, RT is determined by a combination of the time taken to reach the bound and a stimulus-independent non-decision time. As explained in 'Materials and methods', the models in *Figure 2B,C* can be treated as mathematically equivalent. For both depictions, the distance from the starting point of evidence accumulation to the bound, termed the 'excursion', determines how much evidence must be accumulated in order to commit to a choice. In theory, the excursion may change as a function of decision time, for example, if the bound collapses during the course of decision formation. However, for simplicity, we first consider the restricted set of models with static excursions.

Monkeys traded off speed and accuracy by changing the amount of evidence that was needed to commit to a choice. In particular, SAT was explained by the bounded accumulation model with a change in the decision variable excursion. Overall, the excursion was reduced by 43% ± 2% in the high speed regime compared to the high accuracy regime (monkey D: 58% ± 3%; monkey E: 23% ± 4%). Thus, less accumulated evidence was needed to commit to a choice in the high speed regime compared to the high accuracy regime.

The change in SAT was not explained in a consistent way by the other model parameters. Overall, we observed a 33.4 ± 5.8 ms shorter non-decision time in the high speed regime compared to the accuracy regime, and this effect was not consistent between the two monkeys (monkey D: 3.1 ± 5.2 ms

longer in high speed regime; monkey E: 63.7 ± 9.1 ms shorter in high speed regime). Importantly, the RT elongation in the high accuracy regime exceeded any changes observed in non-decision time by a large margin (mean RT elongation, monkey D: 176 ± 4 ms; monkey E: 159 ± 3 ms). Moreover, if changes in RTs were solely a result of altered non-decision time, there would be no associated changes in accuracy. In principle, changes in the SAT could be associated with changes in the signal-to-noise ratio of the sensory evidence, owing perhaps to a change in attentional state, which would lead to a change of sensitivity to motion strength (parameter $k$ of the model). We cannot rule this out, but the model fits provided inconsistent support for such a mechanism in the two monkeys. In the high speed regime, $k$ changed by a factor of 1.36 ± 0.12 and 0.78 ± 0.07, respectively, for monkeys D and E (1.01 ± 0.06 in combined data).

We cannot rule out the possibility that some of the behavioral change was manifested by a competing process in which the monkey simply guessed quickly, so-called 'fast guesses' (*Ratcliff and Rouder, 1998*). A fast guess process could lead to faster and less accurate choices in the high speed regime. However, the level of accuracy for both monkeys at the highest motion strength in the high speed regime suggests that any such fast guess mechanism plays a relatively minor role because fast guesses would reduce accuracy for all stimulus strengths. Together, these considerations show that the change in SAT was achieved primarily via a process that alters the excursion required for the accumulating evidence to support termination.

Traditionally, bounded accumulation models implement this change in excursion by altering the bound height. However, the brain is thought to employ competing accumulator processes for decision making (*Figure 2C*; *Usher and McClelland, 2001*; *Gold and Shadlen, 2007*; *Churchland et al., 2008*). Accordingly, the brain can achieve a reduction in the excursion of accumulated evidence in two ways: (1) by lowering the bounds for both processes, or (2) by adding a signal to the accumulated evidence for both processes. These can be treated as mathematically equivalent: in both cases the effect of the change in the excursion is to alter the amount of accumulated evidence necessary to terminate the process. However, they imply different neural mechanisms.

## Neurophysiology

We recorded neural activity from 70 neurons while the monkeys performed the task (35 in the high speed regime and 35 in the high accuracy regime). In each recording session, we screened neurons based on whether they exhibited spatial selectivity during a delayed saccade task and persistent delay activity during a memory-guided saccade task. For the motion discrimination task, one of the choice targets was placed in the RF of each recorded neuron; the other choice target was placed in the opposite hemifield.

In both speed-accuracy regimes, LIP neurons exhibited signatures of evidence accumulation. These features are illustrated in *Figure 3*, and they are similar to previous observations (*Roitman and Shadlen, 2002*; *Churchland et al., 2008*). Initially, the responses were elevated in the presence of the choice targets, one of which was in the neuron's RF. For the first 200 ms after motion onset, there was no relationship between stimulus strength and firing rate (*Figure 3A,B*; *Equation 1*, p>0.1 for both regimes in each monkey). Onset of the random dot motion (outside the RF) then induced a transient depression in the firing rate that recovered over a 200-ms period. This pattern was similar for speed and accuracy regimes, but the average firing rate in this epoch was 15% ± 1% lower in the high accuracy regime (monkey D: 24% ± 2% lower in the high accuracy regime; monkey E: 8% ± 2% lower in the high accuracy regime; p<0.01 in all cases), as discussed in detail below.

After the transient depression in the firing rate, the average responses began to display characteristic ramping activity that depended on stimulus strength and direction (*Figure 3A,B*). To characterize this relationship, we estimated the 'buildup rate' for each motion strength ('Materials and methods'). Stronger motion in the $T_{in}$ direction led to larger buildup rates, and stronger motion in the $T_{out}$ direction led to smaller buildup rates (*Figure 3C*; *Equation 2*, p<0.01 for each monkey in both regimes). This is consistent with the bounded accumulation framework, which posits that the drift rate of the decision variable should depend on the strength of the evidence. Interestingly, larger buildup rates were observed in the high speed regime (*Figure 3C*; *Equation 3*, p<0.05 for each monkey). Importantly, however, there was not a significant difference between the slope of the relationship between buildup rate and motion strength between the two regimes (*Equation 3*, p>0.1 for each monkey). This suggests that the sensory evidence influences the decision variable in the same way in both speed-accuracy regimes, consistent with the conclusions drawn from the fits to behavior.

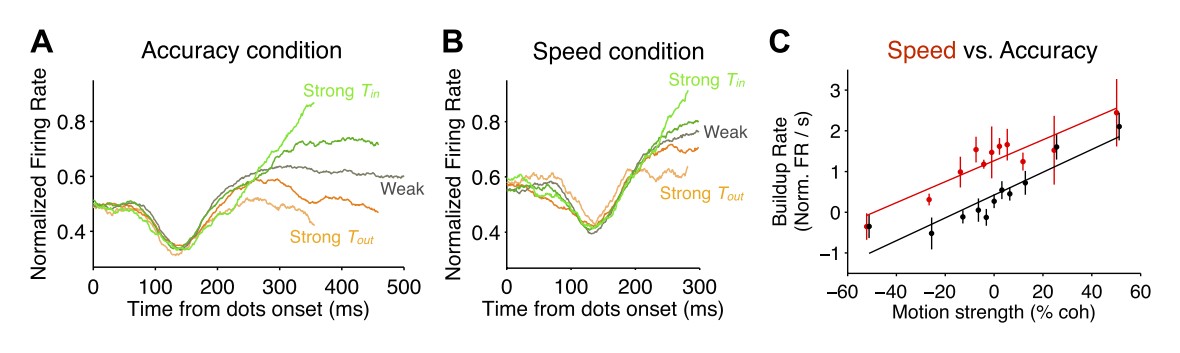

**Figure 3**. Comparison of neural responses accompanying decision formation in the two speed-accuracy regimes. (**A** and **B**) Population firing rates plotted as function of time from stimulus onset and sorted by stimulus strength. Lighter green colors correspond to stronger $T_{in}$ motion; lighter orange colors correspond to stronger $T_{out}$ motion; darker colors correspond to intermediate stimulus strengths. Averages depict combined data from both monkeys. In both regimes, the responses exhibit characteristic ramping profiles that depend on stimulus strength approximately 200 ms after stimulus onset. Prior to averaging, responses for each neuron were normalized based on the peri-saccadic response (0–50 ms prior to saccade initiation) measured during a block of visually-guided delayed saccade trials. N = 35 neurons in each speed-accuracy regime. Error bars show SEM. For each condition, traces end at the point beyond which fewer than 50% of the trials would contribute to the averages, owing to the different RTs for each trial. (**C**) Buildup rate plotted as a function of motion coherence for the high accuracy regime (black) and the high speed regime (red). Error bars show SEM for the estimated buildup rates for each data point. Line fits capture the relationship between buildup rate and motion strength. Overall, the buildup rates were shifted to larger values in the high speed regime, but the change in buildup rate per unit motion strength was similar in the two regimes.

In both speed-accuracy regimes, LIP neurons also reflected a signature of decision commitment. Shortly before saccade initiation for $T_{in}$ choices, rather than reflecting stimulus strength, responses attained a common level of activity (*Figure 4A*; *Equation 1*, p>0.3 for coherence effect 75 ms before the saccade for each monkey in both regimes). The coalescence of $T_{in}$ responses is also evident when responses are grouped by RT rather than stimulus strength (*Figure 4B,C*; *Equation 4*, p>0.2 for RT effect 75 ms before the saccade for each monkey in both regimes). This overall pattern is consistent with the idea that LIP responses support the presence of a decision bound which when reached, terminates further deliberation and triggers commitment to a choice.

Interestingly, the level of this decision bound was undifferentiated by the speed-accuracy regime. Recall that the bounded evidence-accumulation model explained the behavioral data with a change in the excursion of the decision variable in the two regimes. The brain could implement a change in excursion by altering the threshold level of LIP firing rate necessary to commit to a choice. However, we did not observe significant differences between the firing rates in the two regimes within 200 ms of the saccade for either monkey (*Figure 5A*; F test, *Equation 5*, p>0.5 for each monkey).

Alternatively, the excursion could be modified by means of an evidence-independent signal that adds directly to the accumulated evidence. We looked for such a signal by averaging responses in each speed-accuracy regime across all stimulus strengths and both directions to extract the component of the response that was not dependent on the evidence. This evidence-independent component was significantly elevated in the high speed regime relative to the high accuracy regime (*Figure 5A*). In one monkey, this difference began from before motion onset and persisted throughout the period of evidence accumulation (p<0.001). For the other monkey, a significant difference emerged 100 ms after motion onset (p<0.01) and persisted at all time points thereafter (p<0.001). Thus, for both monkeys, LIP neurons undergo a smaller excursion in firing rate from the start to the end of the decision in the high speed regime compared to the high accuracy regime. It appears that an evidence-independent signal effectively pushes the neural representation of accumulated evidence closer to a threshold.

This conclusion receives additional support from an analysis of individual neurons. Across all recorded neurons, there was a significant negative correlation between the estimated evidence-independent neural response and the mean RTs for the corresponding behavioral sessions (*Figure 5B*, r = −0.57, p<0.01). Of course, this correlation is driven to a large extent by the changes in the mean values between the two regimes. Nonetheless, there is variation within each regime, and for neurons in each regime individually, the correlation was significant (r = −0.40 and −0.63 for high speed and high accuracy regimes, respectively; p<0.05 in both cases).

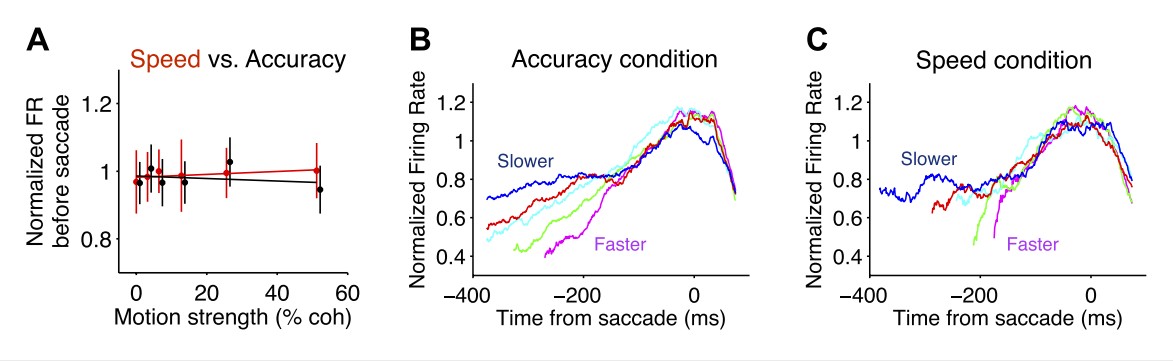

**Figure 4**. Comparison of neural responses accompanying decision termination in the two speed-accuracy regimes. (**A**) Pre-saccadic responses plotted as a function of motion coherence for $T_{in}$ choices. Firing rates were measured in a 50 ms window, centered 75 ms before the saccade (accuracy and speed regime in black and red, respectively). The flat regression is consistent with the idea that the responses reach a common value before $T_{in}$ choices, for all motion strengths. Normalization procedure is identical to *Figure 3*. (**B** and **C**) $T_{in}$ responses plotted as a function of time from the saccade and sorted by reaction-time quintile (colors). Averages depict combined data from both monkeys. In both speed-accuracy regimes, responses are stereotyped before the saccade. N = 35 neurons in each speed-accuracy regime. Error bars show SEM.

Beyond correlation, these changes in the evidence-independent neural signal explained the changes observed in the behavior between the two speed-accuracy regimes. One way to appreciate this is to fit the behavior in both regimes with the bounded accumulation model such that the only difference between the two regimes is established by the measured evidence-independent neural signals. Notice that in both regimes, the evidence-independent signal increased over time (*Figure 5A*), consistent with urgency signals reported previously (*Ditterich, 2006*; *Churchland et al., 2008*; *Cisek et al., 2009*). For each monkey, we approximated the evidence-independent signals in each regime with hyperbolic functions (*Figure 6A*), and used these neurally-derived 'urgency functions' to control the model's excursion for each regime. Importantly, the change in excursion between the two regimes is not fit to the behavioral data; it is measured from the neural data. In this way, the only free parameters in the model are the three that are shared between the two regimes ('Materials and methods'). This neurally-constrained model provided a satisfying account of the magnitude of the changes in performance and RT for the two regimes ($R^2$ = 0.99 for monkey D; $R^2$ = 0.97 for monkey E; *Figure 6B*).

In traditional diffusion models, like the one depicted in *Figure 2B*, the neurally-derived, evidence-independent signal would be represented by a time-dependent change in the height of the bound (*Ditterich, 2006*; *Drugowitsch et al., 2012*). In the high speed regime, the starting point of the bound is nearer to the origin, and the rate of collapse is more severe. In the race-like architecture (*Figure 2C*), a change in bound height is equivalent to a change in the starting point of both accumulations, and a collapsing bound is equivalent to a time-dependent, monotonically increasing signal that is also added to both accumulators. The neural responses are consistent with the idea that both components comprise a singular, evidence-independent signal that adds to the evidence-dependent activity in LIP. Nonetheless, one might ask how well each component alone could explain the behavioral changes between the two speed-accuracy regimes. To test this, we fit the behavioral data using only the baseline or only the time-dependent component of the neurally-derived, evidence-independent signal. The model comparison, summarized in *Table 1*, suggests that the time-dependent component of the signal is more important, although both components contribute to the fits in *Figure 6D*, especially for monkey D. This dissection is far less important than the rather remarkable observation that a pair of signals derived from the physiology in two speed-accuracy regimes and scaled identically, can explain the dramatic differences in the behavior.

## Discussion

The computations that underlie decision making must dictate not just what is decided but also when it is decided. Thus, the brain requires termination rules to end deliberation during a decision process. Just as the criterion in signal detection theory establishes a balance between false alarms

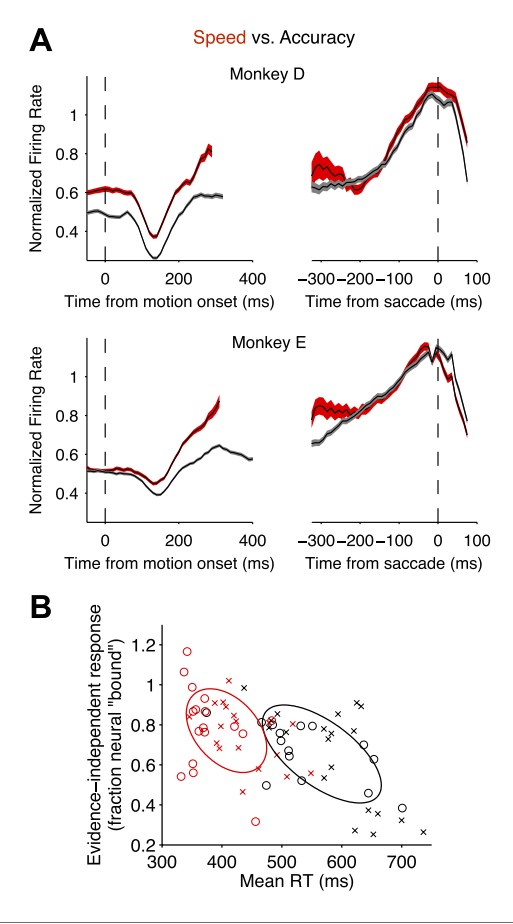

**Figure 5**. Speed-accuracy tradeoff alters LIP responses from the beginning of the decision process. (**A**) Population firing rates plotted for each monkey. Red and black traces correspond to the speed and accuracy regimes, respectively. On the left, responses are plotted as a function of time from motion onset averaged across both directions of motion and all motion strengths (same normalization as in **Figures 3 and 4**). Trials contribute data to the average only up to 100 ms before the saccade. Averages are shown up to the time when half of the 50% coherent motion trials contribute data. On the right, responses are plotted as a function of time from the saccade. Responses are averaged across all trials that ended in a $T_{in}$ choice. Trials contribute data to the average only from 200 ms after motion onset onwards. Curve thickness (shading) shows SEM. N = 17 and 15 neurons for the high speed and high accuracy regimes, respectively, for monkey D. N = 18 and 20 neurons for the high speed and high accuracy regimes, respectively, for monkey E. (**B**) The relationship between the magnitude of the evidence-independent neural response and mean RT across experiments. The response statistic captures the evidence-independent component of the firing rate early in the decision process, expressed as a fraction of the average response from the neuron 75 ms before all $T_{in}$ choices ('Materials and methods'). Larger values

and misses (**Green and Swets, 1966**), a termination rule for a decision process can establish a balance between speed and accuracy. We found that monkeys can control their speed-accuracy policy via a process that changes the amount of accumulated evidence needed to commit to a choice. This is consistent with psychophysical studies in humans that suggest the SAT is implemented as a change in the decision bound (**Reddi and Carpenter, 2000**; **Reddi et al., 2003**; **Palmer et al., 2005**).

In neurophysiology, the bound is conceived as a threshold applied to the firing rates of neurons that represent the accumulated evidence for one or the other alternatives. Although we do not know which neurons in the brain are responsible for applying this threshold, a signature of the process is the stereotyped level of firing rate of LIP neurons just preceding a $T_{in}$ decision. Interestingly, we found that the level of this threshold does not change when the monkey adopts a different speed-accuracy tradeoff. Instead of changing the level of the threshold, the brain changes the level of the starting point of the accumulation and adds a time-dependent signal to the accumulated evidence. The effect is to push the accumulated evidence closer to the fixed threshold. This dynamic, evidence-independent signal has been termed urgency (**Ditterich, 2006**; **Churchland et al., 2008**; **Cisek et al., 2009**; **Drugowitsch et al., 2012**) because it reflects the temporal cost associated with decision formation. It is mathematically equivalent to a collapsing bound because both mechanisms would alter the excursion of accumulated evidence from start to end of the decision, thereby allowing the brain to satisfy a mixture of desiderata related to time and evidence to terminate a decision. At the extremes of the mixture lie a temporal deadline or a fixed level of evidence (e.g., flat bound).

Apart from the change in termination rule, other aspects of the decision making process were unaffected by the speed-accuracy regime. The fits to even the simplest diffusion model (with flat bounds) indicate that the conversion from motion to accumulated evidence (e.g., drift rate) did not change consistently in the two regimes. This is further supported by the neural recordings. As adduced from the slopes of the graphs in **Figure 3C**, a change in motion coherence induces the same change in buildup rate in both regimes. The constant offset between these graphs reflects an evidence-independent component of the neural response—that is, the dynamic component of the urgency signal. This component combined with the change in baseline is mathematically

*Figure 5. Continued*

imply that less evidence-dependent signal is required to reach the bound. A value of 1 indicates that on average the evidence-independent signal alone would reach the 'neural bound' by 300 ms. Neurons from monkey D are shown as circles; neurons from monkey E are shown as crosses. Red symbols correspond to neurons recorded in the high speed regime; black symbols correspond to the high accuracy regime. Ellipses are drawn to the 50% confidence region based on the covariance matrix calculated for each speed-accuracy regime. The correlations in each regime were significant (p<0.05).

equivalent to symmetrically collapsing bounds. Thus we were able to explain the difference in accuracy and RT in the two regimes by fixing all other parameters of the diffusion model to be identical between the two regimes (*Figure 6C*). We simply introduced the urgency signals, measured physiologically, by converting from fraction of the firing rate 'bound' to the units of standard diffusion. The scaling itself (*α*; *Equation 8*) was identical in the two regimes.

Why would the brain add a signal to the accumulated evidence rather than adjusting the threshold? One possibility is that it simplifies the task of the downstream structures responsible for decision termination (*Marshall et al., 2012*), rendering it similar to operations involved in simpler motor control (*Hanes and Schall, 1996*; *Cisek and Kalaska, 2010*). The neural computations involved in sensing a signal threshold crossing must be affected by the fidelity (i.e., signal-to-noise ratio) of the signal, and it is well known that fidelity is greater at higher firing rates (because variance, not standard deviation, scales with mean firing rate). Indeed, it has been argued that some type of temporal smoothing is required to sense a threshold (*Mazurek et al., 2003*; *Heitz and Schall, 2012*). By fixing the threshold to a common level, the mechanism would avoid the need to compensate for these changes.

This architecture is supported by neuroimaging studies, which show an increase in baseline activity in decision-related areas of the human brain when speed is given precedence over accuracy (*Forstmann et al., 2008*; *Ivanoff et al., 2008*; *van Veen et al., 2008*; *Forstmann et al., 2010*; *Wenzlaff et al., 2011*) and with theoretical considerations that favor this implementation (*Marshall et al., 2012*; but see; *Lo and Wang, 2006*). The origin of the urgency signal is unknown, but it may be related to other stimulus-independent signals observed in monkey LIP, which have been shown to represent reward and value (*Platt and Glimcher, 1999*; *Rorie et al., 2010*), prior probability (*Hanks et al., 2011*; *Rao et al., 2012*) and elapsed time (*Leon and Shadlen, 2003*; *Janssen and Shadlen, 2005*). In addition, when faster speeds are cued in a search task, a large stimulus-independent additive signal with a time-dependent component is evident in FEF responses, along with more complex changes in decision threshold (*Heitz and Schall, 2012*; their *Figure 3*).

The tradeoff between speed and accuracy illustrates the flexibility with which the brain can establish policies on decision processes. Control of these policies endows individuals with the ability to reach different choices based on the same information (*Shadlen and Roskies, 2012*). We have revealed one potential neural mechanism by which a policy change can affect a decision process. We believe that studying how such policy control can go awry is an essential step in better understanding disorders of higher brain function, for instance by explaining what leads to more careful or more impulsive decision making when individuals are faced with the same circumstances.

## Materials and methods

### Behavioral task

Two rhesus monkeys (Macaca mulatta) were trained to perform a motion discrimination task. Visual stimuli were presented on a computer monitor (75 Hz frame rate) using the Psychophysics Toolbox for Matlab (*Brainard, 1997*). Trials began with the appearance of a single dot that the monkey was required to fixate. To minimize effects of anticipation on behavior and neural activity, random intervals between trial events were implemented using truncated exponential distributions, $t_{min}$ + exprnd($\mu$), where exprnd is a random sample from an exponential distribution with mean = $\mu$. If the generated interval was longer than the maximum allowed time for the epoch ($t_{max}$), the interval was resampled to ensure it was less than $t_{max}$. After a variable delay ($t_{min}$ = 300 ms, $\mu$ = 300 ms, $t_{max}$ = 1 s), two bright red choice targets appeared at an equal distance from the fixation point and 180° apart. After another variable delay ($t_{min}$ = 300 ms, $\mu$ = 300 ms, $t_{max}$ = 1.5 s), the random dot motion stimulus appeared in an aperture 5° in diameter that was centered at the fixation point. The dynamic random dot stimulus consists of sets of random dots, which are shown for one video frame and then updated 40 ms later (e.g., dots in

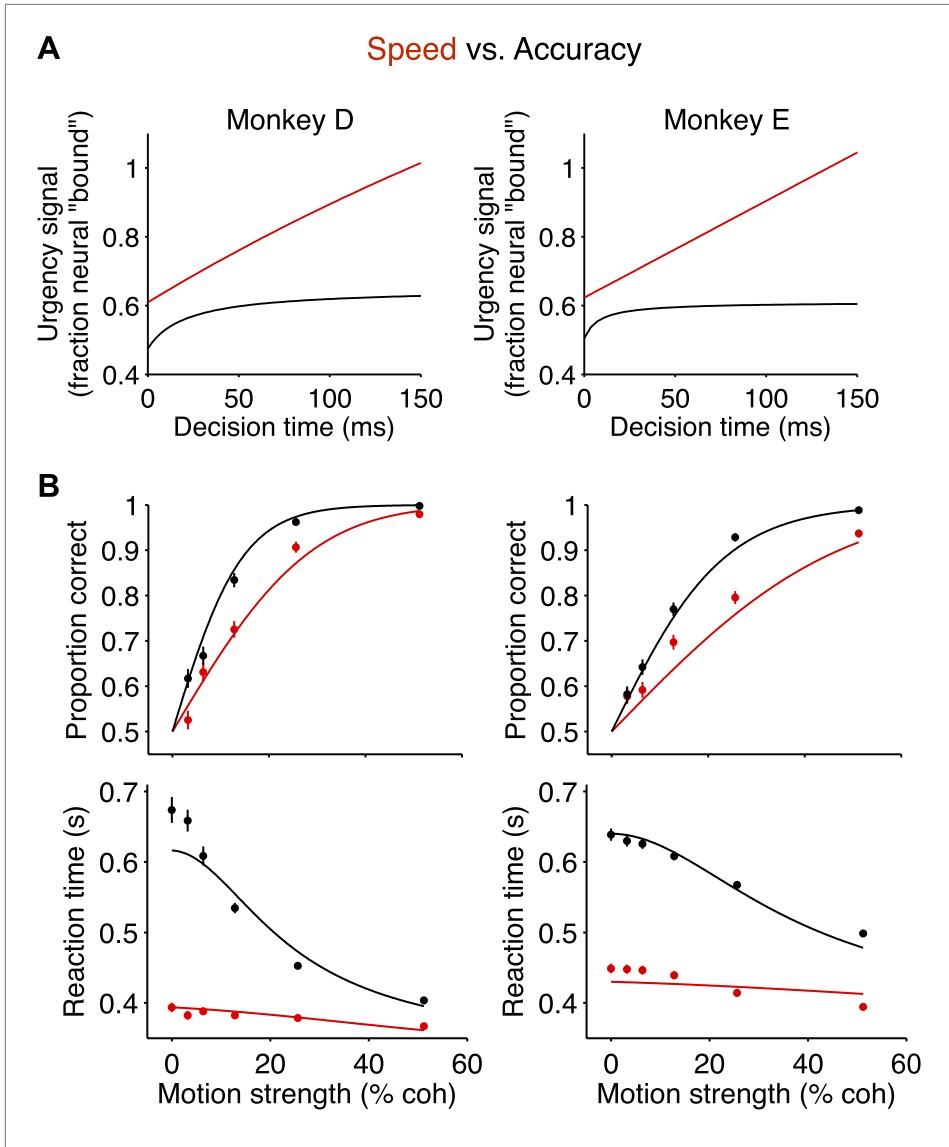

**Figure 6**. The neurally-derived urgency signal explains the magnitude of the behavioral change between speed-accuracy regimes. (**A**) Hyperbolic 'urgency' functions fit to the neural responses from 200 ms after motion onset (decision time = 0) to the point in time beyond which fewer than 50% of the trials contribute to the averages for the highest motion strength in each regime (***Figure 5A***, ***Equation 6*** in 'Materials and methods'). Left column is for monkey D; right column is for monkey E. (**B**) Neurally-constrained fits to behavior. Points show behavioral data from ***Figure 2***. The curves are neurally-constrained model fits to the data. All model parameters are shared (i.e., constrained to be equal) between speed-accuracy regimes except for the difference in excursion, which is measured directly from the neural responses.

---

frame 1 were updated in frame 4; dots in frame 2 were updated in frame 5, and so forth). To update, each dot is either replaced at a random location, or displaced by 14.4 min arc (i.e., 6.0 deg s⁻¹) with probability C (expressed as a percent), termed the motion coherence or motion strength. The dot density was 16.7 dots deg⁻² s⁻¹.

For each trial, there were two possible directions of motion, differing by 180°. For one monkey (monkey E), the directional axis of motion corresponded to that formed by the choice targets. For the second monkey (monkey D), the directional axis of motion always corresponded to the horizontal meridian, but the choice target configuration was as described above, identical to the other monkey. Motion strength (the percentage of coherently moving dots) was chosen randomly from

**Table 1.** Bayes Information Criterion (BIC) comparison of fits to behavioral data based on neurally-constrained models

| Model | monkey D | monkey E |
|---|---|---|
| Baseline only | 664 | 340 |
| Time-dependent only | 124 | 174 |
| Full urgency signal | 71 | 170 |

Because the neurally-derived signals are incorporated in the model via a single scaling factor, all of the fits use the same number of free parameters. Lower BIC values indicate preferred models.

a set $C \in \{0\%, 3.2\%, 6.4\%, 12.8\%, 25.6\%, 51.2\%\}$. The monkey's task was to determine the direction of coherent motion, which it indicated by making a saccade to the appropriate choice target (e.g., rightward target for rightward motion, leftward target for leftward motion). The monkey received a liquid reward for all correct choices, and on a random half of the trials in which 0% coherence motion was displayed. The monkeys received no reward for the incorrect choices.

The monkeys were trained to perform a RT version of the motion discrimination task in two different speed-accuracy regimes: a faster speed (hence, lower accuracy) or higher accuracy (hence, slower speed) one. Each regime was maintained for multiple consecutive daily sessions. For both monkeys, we started with the faster speed regime first, followed by the higher accuracy regime, and then switched back to the faster speed regime, with details as follows. The monkeys were initially trained on a version of the task that used an experimenter-controlled duration for the visual stimulus. Both monkeys achieved stable psychometric thresholds in this version of the task before proceeding to the RT version. In the RT version of the task, the color of the fixation point was changed from red to blue and the monkey was allowed to respond any time after motion onset. Previous training experience revealed a natural tendency of monkeys to respond with short latencies. We exploited this tendency by providing only a small incentive for accuracy; a time out penalty for incorrect choices was added to the standard inter-trial interval to discourage fast errors:

$$t_{ITI} = \begin{cases} t_1 + 7\ exp(-0.5 t_R) & \text{after error} \\ t_1 & \text{otherwise} \end{cases}$$

where $t_{ITI}$ is the inter-trial interval, $t_1$ is a constant, typically 1 s, and $t_R$ is the RT. With this approach, the monkeys established a stable speed-accuracy regime with well-behaved psychometric and chronometric functions. We refer to this as the high speed (low accuracy) regime.

After collecting data from LIP neurons in multiple consecutive daily sessions in this regime, each monkey was switched to a higher accuracy regime. This was achieved through the conditional delay of reward. Specifically, reward was always given for correct choices, but it was delayed relative to motion onset. The duration of this delay was 600–800 ms. So, if the delay was 800 ms and the monkey responded correctly in 500 ms, it would have to wait an additional 300 ms before delivery of the reward. This eliminated any reward-based incentive to respond faster than the minimum delay to reward, encouraging longer deliberation times that would improve accuracy. After multiple daily sessions with this reward structure, the monkeys achieved a new stable speed-accuracy set point involving longer RTs and higher accuracy. Data were collected from LIP neurons in multiple consecutive daily sessions in this regime.

Finally, the monkeys were moved back to the high speed regime. We performed a variety of manipulations to bring the monkeys to a similar speed-accuracy regime as observed in the initial high speed sessions. At first, the monkeys preserved their higher accuracy set point even after removal of the conditional delay of reward for multiple daily sessions. After the additional removal of the time out penalty for incorrect choices, one monkey (monkey E) slowly reached a faster speed set point over the course of 15 sessions. The other monkey (monkey D) required an additional manipulation. For 50% of trials, the motion stimulus was extinguished after 600 ms, so no further sensory information was available. After three sessions with this design, the monkey reached a faster speed set point. The task structure then reverted back to allowing the motion stimulus to be displayed indefinitely, and the monkey maintained the faster speed set point. After stable behavior was achieved in this regime, data were collected from 12 more LIP neurons (7 for monkey D, 5 for monkey E). For both monkeys, the behavior in these final fast speed sessions was similar to the earlier fast speed sessions that were collected before the high accuracy sessions (mean RT difference <35 ms and mean accuracy difference <2.5%). The main effects on neural responses were similar as well, so data were combined across the two sets of high speed sessions.

## Behavioral analyses

Behavioral data were fit using a bounded evidence-accumulation model, also known as bounded drift-diffusion (*Figure 2B*). The model explains choice and decision time as an accumulation of momentary evidence to an upper bound or a lower bound, corresponding to the two direction choices. The momentary evidence gathered in each time step is drawn from a Gaussian distribution with unit variance for 1 s and mean $\mu$ determined by a linear transform of the motion strength: $\mu = kC$, where $C$ is the motion strength and $k$ is a free parameter that scales the motion strength appropriately. The process terminates when the accumulated evidence, termed the decision variable, reaches $\pm B$, the upper and lower decision bounds. The bound reached first by the accumulated evidence determines the choice, and the decision time is determined by how long it takes to reach that bound. RT is a combination of this decision time with an additional non-decision residual time, $t_{nd}$, which accounts for visual and motion delays and other factors that are independent of the decision process. The basic model (fits in *Figure 2*) uses three free parameters: $k$, $B$, and the mean $t_{nd}$ ($\overline{t_{nd}}$).

This model can be described in a mathematically equivalent way in which one process is replaced by two competing processes with perfectly anti-correlated accumulations of moment-by-moment sensory evidence bearing on each alternative (*Figure 2C*; *Usher and McClelland, 2001*). In this version, the choice is dictated by which process first reaches the bound. A signal that adds an equivalent amount to both processes would be equivalent to a reduction of the bound of both processes by the same value. We refer to the distance between the bound and starting point of each process as the 'excursion'. In a single process version of the model, the excursion is controlled entirely by the heights of the decision bounds. A symmetric reduction of the magnitude of each bound is mathematically equivalent to an equally-sized additive signal in the two-process version of the model.

To determine how the model accounts for the difference in behavior between the speed and accuracy regimes, we allowed all three parameters of the model to vary between the two regimes. As is well known, the simplified model, described above, cannot explain mean RT on error trials or the full shapes of the RT distributions for this task (*Ratcliff, 1978*; *Ratcliff and Rouder, 1998*; *Ditterich, 2006*). We therefore maximized the likelihood of the choice proportions (binomial error) and the mean and SD of the RT from correct choices (Gaussian error). Each model's predictions for these values were derived from the analytic solutions for the boundary-crossing probability and first-passage times for a bounded drift-diffusion process (*Palmer et al., 2005*).

## Neural recordings

We recorded from 70 well isolated single neurons in the lateral intraparietal area (LIP) of two monkeys while they performed the reaction time (RT) motion discrimination task. In monkey D, 17 neurons were recorded in the high speed regime and 15 in the high accuracy regime. In monkey E, 18 neurons were recorded in the high speed regime and 20 in the high accuracy regime. Standard methods were used for extracellular recording of action potentials from single neurons (e.g., *Roitman and Shadlen, 2002*).

Neurons were selected using anatomical and physiological criteria. Registration of structural MRI scans to a standard cortical atlas (Caret software; *Van Essen et al., 2001*) was used to identify LIP and to direct the placement of recording electrodes. Registration of recording sites to structural MRI suggests that our neural recordings were obtained from the ventral subdivision of area LIP (LIPv; *Lewis and Van Essen, 2000*). Once the proper anatomic location was identified, we recorded from neurons during the motion task if they met two criteria: (1) spatially selective activity during visually-guided delayed saccade trials at both the time of visual onset and the time of the saccade, and (2) memory activity during the delay period for memory-guided saccade trials (*Gnadt and Andersen, 1988*). Both criteria were assessed qualitatively by the experimenter after finding a neuron with an isolated spike waveform.

In the visually-guided delayed saccade task, each monkey maintained its gaze at a central fixation point while a target was displayed at a peripheral location. The monkey was required to maintain fixation for a delay taken from a truncated exponential as described above ($t_{min}$ = 300 ms, $\mu$ = 600 ms, $t_{max}$ = 2 s). After this delay, the fixation point was extinguished, instructing the monkey to make a saccade to the location of the target. This task was used to find the response field (RF) of each neuron—that is, the region of space where a target would elicit an increased response for the neuron throughout this task. In the memory-guided saccade task, the peripheral target was flashed briefly (150 ms) rather than maintained. Thus, the monkey was required to remember its location during the delay period. After the delay and the instruction to go, the monkey was required to make a saccade to the

remembered location of the flashed target. When collecting neural data while the monkey performed the direction discrimination task, one target was placed in the center of the recorded neuron's RF and the other target was placed in the opposite hemifield. We refer to the target in the neuron's RF and its associated motion as $T_{in}$; we refer to the other target and its associated motion as $T_{out}$.

All training, surgery and experimental procedures were in accordance with the National Institutes of Health Guide for the Care and Use of Laboratory Animals and were approved by the University of Washington Animal Care Committee.

## Analyses of neural data

Population responses for motion discrimination trials were calculated by combining normalized responses across all neurons. Responses for each neuron were normalized by division by the mean peri-saccadic response (0–50 ms prior to saccade initiation) measured during a block of visually-guided delayed saccade trials. The spikes from each trial were aligned with respect to two trial events. Spikes were aligned to motion onset and saccade initiation, respectively, for analyses of the beginning and end of the decision process. When aligning to motion onset, the time period within 100 ms before saccade initiation was excluded to reduce contamination of the averages with peri-saccadic activity. Similarly, when aligning to the saccade, the time period within 200 ms after motion onset was excluded to avoid contamination by the dip in firing rate that follows motion onset. Response averages (running means) were calculated after applying a symmetric (non-causal) boxcar filter to the spike trains (boxcar width = 50 ms), appropriately normalized.

For plotting purposes (*Figure 3A,B*), motion strengths were binned into five groups: strong $T_{in}$ (51.2% coh favoring $T_{in}$), medium $T_{in}$ (25.6, 12.8, and 6.4% coh favoring $T_{in}$), weak (0 and 3.2% coh), medium $T_{out}$ (25.6, 12.8, and 6.4% coh favoring $T_{out}$), and strong $T_{out}$ (51.2% coh favoring $T_{out}$). All statistical tests were performed using the actual motion strengths; they were not influenced by the groupings of the trials. When plotting responses aligned to stimulus end (*Figure 4B,C*), trials were sorted into quintiles based on RT. This was done separately for the responses for each neuron and then combined across corresponding quintiles.

To estimate buildup rate during decision formation (*Figure 3C*), we applied weighted regression to the mean firing rates for each motion strength and direction. The weighting is based on the time-dependent covariance of the running mean (i.e., the sample covariance of the boxcar filtered trials, divided by the number of trials), thereby correcting for temporal correlation induced by filtering and in the LIP responses themselves (*Churchland et al., 2011*). The chosen analysis window was on the initial buildup, beginning 200 ms after motion onset and ending at the point in time when 50% of the trials were still included in the averages for each motion strength in the two speed-accuracy regimes, taking into account the RTs associated with each motion strength in each regime. The value of 200 ms is based on the established latency between stimulus motion (near the fovea) and the emergence of signals in LIP neurons that represent accumulation of evidence in favor of peripheral choice targets in the RFs of the neurons (*Roitman and Shadlen, 2002*; *Huk and Shadlen, 2005*; *Kiani et al., 2008*; *Churchland et al., 2011*), and it is supported in the current study by the time of the earliest significant effects of stimulus strength on LIP responses. We chose an ending point that equated the attrition of trials from each of the averages—that is, the fraction of trials that contribute to the averages. Attrition of trials associated with the fastest RTs biases the estimate of the buildup rate toward shallower slope because the faster rise times drop out of the average. An alternative approach would be to estimate the buildup rate using identical time intervals in both regimes, but Monte Carlo simulations revealed that it would result in a larger underestimation in the high speed regime than using an approach that based the endpoint on a matched level of trial attrition (results not shown). The latter approach does not eliminate the bias entirely, but it minimizes the difference in the bias of the buildup rate estimates in the two regimes.

For the single neuron analysis (*Figure 5B*), we derived estimates of the evidence-independent response for each neuron and the mean RT for the corresponding behavioral session. Because of the noise inherent in measurements from single neurons, we estimated the evidence-independent signal by averaging responses across all motion strengths and choices. For the same reasons as described above, for each neuron we fit a line to the responses in an interval beginning 200 ms after stimulus onset and ending at a point in time when 50% of the trials for highest motion strength contributed to the average, thereby ensuring inclusion of at least 50% of trials at all time points for all motion strengths. From the linear fits, we obtained the firing rate 300 ms after motion

onset (i.e., reflecting 100 ms of processing time in LIP) and divided this by the mean response 75 ms (using 50 ms bin width) before saccades to $T_{in}$. This yields an estimate of the urgency signal in units of 'fraction of neural bound' for each neuron. The scatter plot (*Figure 5B*) displays the relationship between this statistic and the mean RT for each of the 70 experiments. We report standard correlation coefficients and report significance based on the Fisher z-transformation.

Statistical comparisons of neural measures were based on regression analyses using weighted least squares, which respect the standard error for point estimates and their correlation, when applicable. To test the effect of motion strength on firing rate during a specified epoch for each speed-accuracy regime, we fit the model:

$$y_{nfr} = \beta_0 + \beta_1 C \qquad (1)$$

where $y_{nfr}$ is the mean normalized firing rate for each motion strength during the measured epoch, $C$ is the motion strength, and for all equations, $\beta_i$ are fitted regression coefficients. The null hypothesis is that motion strength does not affect firing rate ($H_0$: $\beta_1 = 0$). To test the effect of motion strength on the buildup rate of the neural responses for each speed-accuracy regime, we fit the model:

$$y_{bu} = \beta_0 + \beta_1 C \qquad (2)$$

where $y_{bu}$ is the buildup rate at each motion strength (see above) and C is the motion strength. The null hypothesis is that motion strength does not affect buildup rate ($H_0$: $\beta_1 = 0$). To test whether motion strength influenced the buildup rate differently between the two speed-accuracy regimes, we fit the expanded model:

$$y_{bu} = \beta_0 + \beta_1 C + \beta_2 I_{SA} + \beta_3 I_{SA} C \qquad (3)$$

where $y_{bu}$ is the buildup rate at each motion strength (see above), $C$ is the motion strength, and $I_{SA}$ is an indicator variable for the speed-accuracy regime (1 for speed, 0 for accuracy). The null hypothesis that the regime does not affect the offset of the buildup rate is: $H_0$: $\beta_2 = 0$. The null hypothesis that the regime does not affect the relationship between buildup rate and motion strength is: $H_0$: $\beta_3 = 0$.

To test whether firing rate depended on RT during a particular epoch for each speed-accuracy regime, we binned trials into RT quintiles for each individual neuron. Then, we combined responses across each RT quintile for each speed-accuracy regime and fit the model:

$$y_{nfr} = \beta_0 + \beta_1 T_{RT} \qquad (4)$$

where $y_{nfr}$ is the normalized firing rate for each RT quintile and $T_{RT}$ is the mean RT across all neurons for trials that fall into the corresponding quintile. The null hypothesis that RT does not affect firing rate is: $H_0$: $\beta_1 = 0$.

To compare mean responses derived from particular epochs in the two regimes, we used either a *t* test or weighted regression to detrend the responses. For example, the difference in mean firing rate in the 200 ms epoch preceding the saccade is captured by the following equation,

$$y = \beta_0 + \beta_1 t + \beta_2 I_{SA} + \beta_3 I_{SA} t \qquad (5)$$

where y is the running mean normalized firing rate. The time-dependent terms achieve detrending. The null hypothesis, $H_0$: $\beta_2 = 0$, is evaluated by an F statistic using weighted regression that incorporates the covariance of the running sample means. For both monkeys, $\beta_3$ was not significantly different from zero, and removal of this interaction term had no effect on the conclusions.

## Neurally-constrained modeling of behavior

To make neurally-constrained fits of the behavioral data, we measured the change in neural excursion in the two speed-accuracy regimes, as follows. We fit the evidence-independent neural responses (urgency signals derived from responses combined across motion strength and choice) from 200 ms after motion onset (the start of decision-related ramping activity) to the time point beyond which fewer than 50% of the trials for the highest motion strength contributed to the averages (same reasons as explained above). This fit was based on the following hyperbolic function:

$$y_{nfr} = r_0 + r_\infty \frac{t}{t + t_{1/2}} \qquad (6)$$

where $y_{nfr}$ is the measured population neural response for each speed-accuracy regime. The parameter corresponds to the neural response at decision time $t = 0$ (200 ms after motion onset). $r_0+r_\infty$ corresponds to the asymptotic value of the urgency signal. The parameter $t_{1/2}$ is a rate parameter that controls how fast the urgency signal approaches its asymptote. We measured the difference between this urgency signal (which begins at $r_0$ at $t = 0$) and $r_b$, the neural activity at the time of putative decision commitment (75 ms before the saccade [*Gold and Shadlen, 2007*]), to determine the neurally-measured excursion as a function of time for each speed-accuracy regime. Thus, the neurally-measured excursion is described by the following equation:

$$E_i(t) = r_{b,i} - r_{0,i} - r_{\infty,i}\frac{t}{t + t_{1/2,i}} \qquad (7)$$

where the subscript $i$ indexes each of the two regimes.

The pair of neural excursion functions derived from the two speed-accuracy regimes can be used to constrain the excursion in the bounded accumulation model. In the model, the excursion is determined by the height of the decision bound, so we convert the time-dependent excursion $E(t)$ to a time-dependent bound $B(t)$. $E(t)$ is in units of normalized neural firing rate and the model's $B(t)$ is in units of the standard deviation of a Wiener process (see *Palmer et al., 2005*; *Shadlen et al., 2006*). To achieve the conversion, we apply an identical scale factor, $\alpha$, to the urgency functions from the two regimes. The bound is thus described by

$$B_i(t) = \alpha E_i(t) \qquad (8)$$

where the subscript $i$ indexes each of the two regimes. The bounded accumulation model was then fit to the choice proportions and the mean RTs from correct choices in both regimes with three free parameters that are all shared: $\alpha$ as described above, and $k$ and $\overline{t_{nd}}$ as described previously. Because there is not an analytic solution for drift-diffusion with time-varying bounds, we solved for mean RTs and accuracy numerically (i.e., propagating the probability distributions of accumulated evidence to the absorption bounds). Importantly, the only differences in the model between the two regimes are the parameters in *Equation 7* that are directly estimated from the neural responses. Thus, the neurally-constrained fitting procedure allows us to assess whether the measured differences in neural activity can account for the behavioral changes associated with SAT without any additional free parameters that are different between the two regimes.

We also compared these fits to neurally constrained fits employing only the static or the dynamic component of $E(t)$: $B_i(t) = \alpha E_i(0)$ and $B_i(t) = \alpha[E_i(t) - E_i(0)]$, respectively. These models have the same degrees of freedom as the full model (i.e., $\alpha$, $k$, $\overline{t_{nd}}$). Comparisons were based on the Bayes Information Criterion (BIC). Models with lower BIC values are preferred over alternatives with higher BIC values.

## Acknowledgements

We thank D Wolpert for helpful advice, C Fetsch, N Odean, Shushruth, and L Woloszyn for comments, and A Boulet and K Ahl for technical assistance.

## Additional information

### Funding

| Funder | Grant reference number | Author |
| --- | --- | --- |
| Howard Hughes Medical Institute | | Michael N Shadlen |
| National Eye Institute | EY11378 | Michael N Shadlen |
| Human Frontier Science Program | RGP0067/2011 | Michael N Shadlen |

The funder had no role in study design, data collection and interpretation, or the decision to submit the work for publication.

### Author contributions

TH, Conception and design, Acquisition of data, Analysis and interpretation of data, Drafting or revising the article; RK, MNS, Conception and design, Analysis and interpretation of data, Drafting or revising the article

## Ethics

Animal experimentation: This study was performed in strict accordance with the recommendations in the Guide for the Care and Use of Laboratory Animals of the National Institutes of Health. All of the animals were handled according to approved institutional animal care and use committee (IACUC) protocols (#2896-01) of the University of Washington. All surgery was performed under isoflurane anesthesia and analgesics to avoid pain and distress.

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
