## [Decision Letter]

Thank you for sending your work entitled “A neural mechanism of speed-accuracy tradeoff in macaque area LIP” for consideration at *eLife*. Your article has been favorably evaluated by a Senior editor and 3 reviewers, one of whom is a member of our Board of Reviewing Editors.

The Reviewing editor and the other reviewers discussed their comments before we reached this decision, and the Reviewing editor has assembled the following comments to help you prepare a revised submission.

This study investigates how neural activity changes when subjects make perceptual decisions (and corresponding choices) when either accuracy or speed are emphasized. The results are interesting and novel in that they provide insight about what exactly changes (in terms of underlying neuronal circuits) when speed and accuracy are traded against each other: a phenomenon that occurs under a wide variety of conditions. There are a few concerns regarding the data analysis and presentation that must be dealt with.

1) The presentation of both behavioral and neural data is rather telegraphic. In most figures (e.g., Figure 3, Figure 4) lines are shown without any details about error bars and number of neurons included in the averaging. In addition to unpacking their data, the authors should also show results (both behavior and neural activity) and perform statistics separately for each animal. It would also be nice to see some example neuron responses, as well as some statistics on individual neurons.

2) One important question is whether the effect of the sensory evidence on the decision is the same in the two conditions. The authors conclude that indeed it is, based on the “similar buildup rates observed in both SA regimes”. However, from the plots in Figure 3b, it appears that the separation between in and out responses for a given coherence is much stronger in the accuracy condition. The slope calculation appears neither robust nor sensitive enough. First, it must be crucially dependent on the chosen interval (200 to 350 ms after motion onset): if the buildup rates were calculated starting from the lowest point of the dip (around 160 ms), all the results would be higher in the speed condition than in the accuracy condition. The chosen interval seems arbitrary, and an objective way must be found.

Furthermore, one reviewer was not convinced that the slope calculation is appropriate to begin with, given that (1) it will partly reflect the common 'urgency' signal, which is different in the two cases (Figure 5, inset), and (2) the response trajectories toward threshold are obviously not straight. It seems that the difference between in and out responses at a given point in time would provide a much more robust measure of the impact of the motion stimulus, and by that measure, there would likely be a significant difference between conditions. Here is another way to think about it: given that there is an 'urgency' signal, a component of the activity that rises independently of the sensory evidence, how strongly do the neural responses *deviate* from it as a function of coherence? To assess the impact of the sensory information regardless of urgency, it is those deviations that should be compared across conditions.

3) The setup for the experiment is to test the idea that a change in the bound, or response threshold, is what accounts for the speed-accuracy tradeoff. This, however, is left somewhat unresolved. The bounded accumulation model required a 43% reduction in the excursion from baseline to threshold to account for the full tradeoff. In contrast, the authors observed an increase in baseline that seems (Figure 5) much smaller (maybe 10%?), but once they discuss the rising urgency signal, they do not touch again on the original mechanism. Thus, the question remains, how much of the tradeoff is explained by the observed difference in excursion? There are two components to the increasing 'urgency' signal, an offset and a time-dependent component. In the model, it is possible to fit them either independently or simultaneously (i.e., to both conditions). So, how much of the effect is captured if just the offset (r0) varies across conditions (assuming a time-dependent component that is the same across conditions), and vice versa, how much of the effect is captured if the offset remains constant and only the time-dependent component is allowed to vary across conditions? This would provide a pretty good, quantitative indication of the importance of each term, and in particular, would resolve more quantitatively the original question implied in the Introduction: how well can a pure reduction in the excursion from baseline to threshold explain the observed tradeoff? Judging from the inset in Figure 5, the temporally varying component of the urgency signal seems much stronger than the offset, but really what matters is how well each component can account for the data.

4) Related to this, in the Discussion, the two components, offset and temporally varying, are lumped together as part of a generic 'urgency' signal. This is a bit confusing. First of all, they may correspond to different neural mechanisms. The results of [18], for instance, show both types of changes, but different neuron types reflected each one to different extents. Second, the proposal by Bogacz et al. and the first model implemented in the manuscript deal just with the offset, whereas Cisek's model considers only the temporally-varying component. So, there are both theoretical and empirical reasons for making a distinction. The authors found that both components changed in their experiment, but still, it would be a good idea to discuss them as separate entities.

---

## [Author Response]

We are grateful to the editor and referees for many suggestions and constructive criticisms. As detailed below, we have revised the paper to incorporate nearly all of them. The major changes to the paper include (i) new analyses by monkey (Figures 2, 5 and 6), (ii) a new single neuron analysis (Figure 5), and (iii) improved and better justified analysis window for buildup rates. We believe these and other changes have improved the logical flow of the paper, hence its readability. The process was also illuminating for us because we verified that the key conclusions were robust to a variety of choices we made to perform the analyses (new and old).

*1) The presentation of both behavioral and neural data is rather telegraphic. In most figures (e.g.,*
Figure 3*,*
Figure 4*) lines are shown without any details about error bars and number of neurons included in the averaging. In addition to unpacking their data, the authors should also show results (both behavior and neural activity) and perform statistics separately for each animal. It would also be nice to see some example neuron responses, as well as some statistics on individual neurons*.

We have taken multiple steps to address these concerns. (A) We have improved the presentation of all statistics, numbers of neurons, and other critical information. We have improved the PSTHs. Instead of displaying mean responses connected by line segments, we show the running means of firing rates and enforce a more sensible rule for averaging as a function of time that we apply consistently in all graphs and analyses. The rationale for this change is explained in detail in response to item 2, below. The running means convey the variation in a direct way, and they have the virtue of replacing the line segments with actual data. (B) We now present results for each animal on the key analyses of the behavior and neural changes accompanying a change in speed-accuracy regime (Figures 2, 5 and 6). (C) We have included a new analysis in the manuscript that displays the effects for individual neurons. The new scatter plot (Figure 5) shows an association between variation of SAT from session to session and variation in the neural activity (the neural “urgency signal”) in the corresponding session. However, because neurons were recorded in only one or the other regime (speed or accuracy), individual example neurons would not be informative about changes between conditions.

*2) One important question is whether the effect of the sensory evidence on the decision is the same in the two conditions. The authors conclude that indeed it is, based on the “similar buildup rates observed in both SA regimes”. However, from the plots in*
Figure 3*, it appears that the separation between in and out responses for a given coherence is much stronger in the accuracy condition. The slope calculation appears neither robust nor sensitive enough. First, it must be crucially dependent on the chosen interval (200 to 350 ms after motion onset): if the buildup rates were calculated starting from the lowest point of the dip (around 160 ms), all the results would be higher in the speed condition than in the accuracy condition. The chosen interval seems arbitrary, and an objective way must be found*.

While our choice of interval for calculating the buildup rate was not arbitrary, we agree that we failed to adequately justify it in the original submission. The starting point, 200 ms, is based on the established latency between stimulus motion (near the fovea) and decision-related activity in LIP neurons that represent the choice targets. This surprisingly long latency has been documented in several publications (e.g., Roitman et al. 2002; [19]; [3]), and it is supported in the current study by the time of the earliest significant effects of stimulus strength on LIP responses. The ending point was guided by trial attrition. At longer times, progressively fewer trials contribute to the averages. In the revised manuscript, we apply a criterion of 50% attrition for each coherence in each SA regime; that is, we stop when fewer than half of the trials would contribute to the response. This is slightly different from what we did in the original submission, and we are grateful to the reviewers for pushing us to consider this issue more carefully. As the reviewers are aware, attrition of trials associated with the fastest RT biases the estimate of the buildup rate toward shallower slope (because the faster rise times drop out of the average). We used Monte Carlo methods to explore the bias introduced by a variety of attrition rules. We had originally chosen an endpoint for the regression to match the value of 50% attrition in the high speed regime, but we discovered that this strategy biased the estimates differently in the two SA regimes, as the reviewer suspected. We now apply the same 50% attrition rule to each motion strength in each of the SA regimes. This helps to clarify the relationship between the PSTHs in Figure 3 and the buildup rates shown in the new Figure 3. Like the original figure, the new Figure 3 shows that a change in motion strength affects the buildup rate similarly in the two regimes (compare the slopes). However, the new graph better clarifies a consistent difference in the absolute magnitude of the buildup rates in the two regimes (compare the offsets), which is consistent with the presence of an urgency signal that affects all buildup rates by roughly the same amount, albeit more so in the high speed regime. We have revised the manuscript to explain this objective basis for the comparisons of buildup rates between the two conditions.

*Furthermore, one reviewer was not convinced that the slope calculation is appropriate to begin with, given that (1) it will partly reflect the common 'urgency' signal, which is different in the two cases (*Figure 5*, inset), and (2) the response trajectories toward threshold are obviously not straight. It seems that the difference between in and out responses at a given point in time would provide a much more robust measure of the impact of the motion stimulus, and by that measure, there would likely be a significant difference between conditions. Here is another way to think about it: given that there is an 'urgency' signal, a component of the activity that rises independently of the sensory evidence, how strongly do the neural responses* deviate *from it as a function of coherence? To assess the impact of the sensory information regardless of urgency, it is those deviations that should be compared across conditions*.

With regard to point (1), we agree with the reviewer that differences in urgency could affect the buildup rate calculation. Importantly, though, urgency changes that underlie the SAT should not depend on motion strength. So, it should not affect the slope of the relationship between buildup rate and motion strength, as shown in Figure 3. We realize that we did not explain this well in the original manuscript, and we have revised the text to clarify this point. With regard to point (2), one of the main reasons that response averages are not straight is the attrition of the trials with more extreme buildup rates from the averages. Comparison of the responses at a particular point in time would suffer from the concerns described above; greater trial attrition in the high speed regime compared to the high accuracy regime at matched times. All of these concerns prompted us to improve (and clarify) the interval chosen for analyses, which is based on matched trial attrition in the two SA regimes.

*3) The setup for the experiment is to test the idea that a change in the bound, or response threshold, is what accounts for the speed-accuracy tradeoff. This, however, is left somewhat unresolved. The bounded accumulation model required a 43% reduction in the excursion from baseline to threshold to account for the full tradeoff. In contrast, the authors observed an increase in baseline that seems (*Figure 5*) much smaller (maybe 10%?), but once they discuss the rising urgency signal, they do not touch again on the original mechanism. Thus, the question remains, how much of the tradeoff is explained by the observed difference in excursion? There are two components to the increasing 'urgency' signal, an offset and a time-dependent component. In the model, it is possible to fit them either independently or simultaneously (i.e., to both conditions). So, how much of the effect is captured if just the offset (r0) varies across conditions (assuming a time-dependent component that is the same across conditions), and vice versa, how much of the effect is captured if the offset remains constant and only the time-dependent component is allowed to vary across conditions? This would provide a pretty good, quantitative indication of the importance of each term, and in particular, would resolve more quantitatively the original question implied in the Introduction: how well can a pure reduction in the excursion from baseline to threshold explain the observed tradeoff? Judging from the inset in*
Figure 5*, the temporally-varying component of the urgency signal seems much stronger than the offset, but really what matters is how well each component can account for the data*.

The reviewers correctly point out a tension between our initial presentation of the behavioral data and the neural responses. Specifically, we show that the bounded accumulation model accounts for the behavioral changes with a large static change in the excursion from baseline to threshold. In contrast, the neural responses exhibit a combination of static and time-dependent components of an evidence-independent signal. However, the reason for this tension is straightforward: in the initial presentation of the model fits to behavioral data, we opted not to include a time-dependent component for the sake of simplicity. We did not intend this to motivate a test of whether a static component alone accounts for the SAT. We view the key finding as the fact that an evidence-independent signal is added to LIP responses to implement the SAT. However, we also acknowledge that this signal can be broken down into its static and time-dependent components. Because some readers may react like the reviewers, and because we wish to retain the simplicity of the fits in Figure 1, we now report how well either the static or dynamic components of the excursion can account for the behavioral data. We hope the additional attention given to this issue throughout the paper will help to clarify our interpretation.

*4) Related to this, in the Discussion, the two components, offset and temporally varying, are lumped together as part of a generic 'urgency' signal. This is a bit confusing. First of all, they may correspond to different neural mechanisms. The results of*
[18]*, for instance, show both types of changes, but different neuron types reflected each one to different extents. Second, the proposal by Bogacz et al. and the first model implemented in the manuscript deal just with the offset, whereas Cisek's model considers only the temporally-varying component. So, there are both theoretical and empirical reasons for making a distinction. The authors found that both components changed in their experiment, but still, it would be a good idea to discuss them as separate entities*.

As the reviewers note, we grouped both of these components together in the Discussion of the original manuscript. The reason we chose to do that is that we have little reason to believe that they are different mechanisms. If there is a mechanism to add a time-dependent signal to a response, it seems unlikely that the mechanism would not extend to t=0. Indeed, we believe that the distinction between static and dynamic components is artificial: the static component is nothing more than the value of the dynamic signal at t=0. As the reviewers recognize, there is a tradition that emphasizes only stationary (i.e., flat) bounds, but this turns out to have been a misguided application of the sequential probability ratio test to situations where it is not normative;in particular situations like ours when there are a variety of stimulus strengths that are randomly interleaved. Of course our lab has contributed to the confusion, until recently (see [8], and Box 2 of Shadlen and Kiani, 2013). Nevertheless, we recognize that we did not make these points clear in the original manuscript. Thus, we have elaborated on these issues in the revised manuscript.